# Local probe-induced structural isomerization in a one-dimensional molecular array

Shigeki Kawai [1,2] ✉, Orlando J. Silveira[3], Lauri Kurki [3], Zhangyu Yuan[1,2], Tomohiko Nishiuchi [4,5], Takuya Kodama [4,5], Kewei Sun [1], Oscar Custance [1], Jose L. Lado [3], Takashi Kubo [4,5] ✉ & Adam S. Foster [3,6] ✉

Synthesis of one-dimensional molecular arrays with tailored stereoisomers is challenging yet has great potential for application in molecular opto-, electronic- and magnetic-devices, where the local array structure plays a decisive role in the functional properties. Here, we demonstrate the construction and characterization of dehydroazulene isomer and diradical units in three-dimensional organometallic compounds on Ag(111) with a combination of low-temperature scanning tunneling microscopy and density functional theory calculations. Tip-induced voltage pulses firstly result in the formation of a diradical species via successive homolytic fission of two C-Br bonds in the naphthyl groups, which are subsequently transformed into chiral dehydroazulene moieties. The delicate balance of the reaction rates among the diradical and two stereoisomers, arising from an in-line configuration of tip and molecular unit, allows directional azulene-to-azulene and azulene-to-diradical local probe structural isomerization in a controlled manner. Furthermore, our theoretical calculations suggest that the diradical moiety hosts an open-shell singlet with antiferromagnetic coupling between the unpaired electrons, which can undergo an inelastic spin transition of 91 meV to the ferromagnetically coupled triplet state.

Atomic force microscopy and scanning tunnelling microscopy (STM), operating at low temperature under ultrahigh vacuum conditions, are powerful tools to investigate single molecules down to the atomic scale[1]. The combination of bond-resolved imaging with a CO terminated tip[2,3] and structural isomerization via homolytic fission[4–9], in particular, opened the field of local probe chemistry. In such studies, halo-substituted planar molecules are adsorbed on surfaces and subsequently the C-X bonds (where, X=Cl, Br, I) are cleaved by the tunneling current flowing from the tip. However, a radical species directly adsorbed on a metal surface is very short-lived, immediately stabilizing by connecting to surface atoms. A common method to electronic

decouple molecules from the metal substrate is by an NaCl ultrathin film[10–12], and this also can be used to prevent the stabilization of the radical and allows for the realization of a non-short-lived product at low temperature[4–9](Fig. 1a, c). By connecting two radicals through tip-induced manipulations, dimers have been successfully synthesized[13]. This bond manipulation process can also be repeated for dihalogenated molecules, resulting in a highly unstable diradical species (Fig. 1)[4,5,8,9].

An alternative approach to an insulating layer on the metal, is to use three-dimensional (3D) hydrocarbons, in which a large gap between the outer group and the metal substrate also prevents the

[1]Center for Basic Research on Materials, National Institute for Materials Science, Tsukuba, Ibaraki, Japan. [2]Graduate School of Pure and Applied Sciences, University of Tsukuba, Tsukuba, Japan. [3]Department of Applied Physics, Aalto University, Helsinki, Finland. [4]Department of Chemistry, Graduate School of Science, Osaka University, Toyonaka, Japan. [5]Innovative Catalysis Science Division (ICS), Institute for Open and Transdisciplinary Research Initiatives (OTRI), Osaka University, Suita, Osaka, Japan. [6]WPI Nano Life Science Institute (WPI-NanoLSI), Kanazawa University, Kakuma- machi, Kanazawa, Japan. ✉e-mail: KAWAI.Shigeki@nims.go.jp; kubo@chem.sci.osaka-u.ac.jp; adam.foster@aalto.fi

**Fig. 1 | Local probe rearrangements of small aromatic compounds. a** Tip-induced syntheses of single aryne[4], **b** 3,4-benzocyclodeca-3,7,9-triene-1,5-diyne[5], and **c** Songheimer-Wong diyne molecules[8]. **d** Tip-induced syntheses of diradical **2** and dehydrozulene **3** units in one-dimensional molecular array.

stabilization of the radical by the underlying metal surface even under the absence of a decoupling layer[14]. Furthermore, since this system offers an in-line configuration between the tip and the outer group of the molecule, the probe can directly sense hydrogen[15,16] and halogen bonds[17]. When a tip terminated by either a bromine atom or a fullerene molecule is brought into proximity to the unpaired electron species pointing out from the surface, a molecule-by-molecule additional reaction can also be conducted[14]. This 3D configuration is also beneficial to investigate the electronic property of the radical species and their isomerization.

Here, we measure electronic and magnetic properties of radical species in 3D organometallic compounds (OMCs). The C-Br bond pointing out from the surface is studied in detail by a combination of low-temperature STM and density functional theory (DFT) calculations. The electronic properties of radical species obtained by the sequential removal of bromine atoms using the tunneling current were investigated by scanning tunneling spectroscopy (STS). Since the diradical species are energetically unstable under debromination conditions, isomerization to dehydroazulene is immediately caused. Tuning the reaction rate by the tip-sample gap, we found that the array unit can be switched between three configurations of the diradical and two dehydroazulenes by the local probe in a controlled manner (Fig. 1c).

## Results and Discussion

### On-surface synthesis of 3D-OMC and electronic properties

We employed 3D-OMC for local probe isomerization (Fig. 2a)[14,18]. Hexabromo-substituted trinaphtho[3.3.3]propellane (6Br-TNP) molecules were in situ deposited on clean Ag(111) surfaces under ultra-high vacuum conditions and subsequently annealed to synthesize 3D-OMC (Fig. 2b). Most of the out-of-plane C-Br bonds in the product remained intact because no catalysis from the surface metal was induced (Fig. 2c). When the bias voltage was set higher than 2.5 V, the targeted C-Br bond was cleaved by the tunneling current. This process was highly reproducible, as a series of bromine atoms were removed one by one in a controlled manner (Fig. 2d and Supplementary Figure 1)[14];

the remaining darker ovals indicate the absence of the bromine atoms. Since the naphthyl group missing two bromine atoms is fairly distant from the metal substrate, we assumed at first that the product should be a diradical. From now on we will discuss the electronic properties of the naphthyl group before and after debromination and its consequential effects.

We first measured the electronic property of the C-Br moiety in the 3D-OMC with STS before the debromination (Fig. 2e) and found that highest occupied molecular orbital (HOMO) and lowest unoccupied molecular orbital (LUMO) levels are located at -1.7 V and 1.8 V, respectively. Note that the debromination was caused by setting the bias voltage above the LUMO level (2.5 eV). This is consistent with other studies exploring the mechanism of debromination in detail[12]. Next, we measured the electronic property of the fully-debrominated unit and found significant energy shifts depending on the measured sites (Fig. 2f). The constant height d$I$/d$V$ maps taken at the LUMO and HOMO levels show distinct contrasts between the C-Br bond and its debrominated sites. While the measured d$I$/d$V$ map shows a textbook-like symmetric contrast on the C-Br bond site due to the π states (Fig. 2g), an asymmetric contrast appeared on the debrominated site revealing the lower part brighter than the upper part (Fig. 2h and Supplementary Figure 2). In order to understand the observed difference in contrast before and after debromination, we conducted DFT calculations considering the ribbon structure on top of an Ag(111) surface (see Methods and Supplementary Fig. 3). The simulated density of states (DOS) map of the dibrominated site is in good agreement with the experimental data (Fig. 2i). However, the STM simulated images of the diradical **2** reveal two symmetrical bright spots at both negative and positive biases, which is inconsistent with our experimental findings and an indication that the diradical **2** is not the final product of the debromination (Supplementary Fig. 4). To account for this inconsistency, we considered a large variety of possible atomic structures as alternatives to the diradical **2**, including the role of Br and H terminations, and isomerization. We found an excellent agreement for the dehydroazulene structure **3** (Fig. 2j) and propose that the asymmetric structure corresponds to the dehydroazulene group formed by the rearrangement of the naphthyl group[19,20]—this structure is about 0.5 eV lower in energy than **2**. The analysis of the calculated DOS (Fig. 2k, l) shows a sufficient decoupling of the imaged states from the Ag(111) substrate, with a clear gap and no states between the HOMO (-1.0 eV) and LUMO (0.5 eV), in agreement with the experimental spectra. The removal of Br results in the dominant states seen in simulated images (Fig. 2j), corresponding to the seven-membered ring (lower part) in the dehydroazulene moiety with a weaker signature of the five-membered ring (upper part) seen in the image at -1.8 V (both rings are seen for negative bias and only the seven membered for positive bias). The calculated molecular orbitals in vacuum have similar features to the electron-rich part at the 1-, 8-, and 10-positions of dehydroazulene (Fig. 1) producing a clover-like HOMO, whereas the electron deficient part at the 8- and 9-positions produces a cotyledon-like LUMO (Fig. 2m, n). This consistency indicates a small interaction between the dehydroazulene group and the underlying metal substrate although the three-fold propellane core is strongly distorted by the adsorption to the metal substrate[15].

### Local probe-induced structural isomerization

We found that applying a large bias voltage induces a reversible chiral switch of the dehydroazulene unit. The tip was first positioned above the seven-membered ring (bright site, as indicated by a red cross in Fig. 3a) and subsequently the bias voltage was swept to a more negative value from zero. We detected abrupt changes in the tunneling current at an applied bias voltage of −2.3 V (Fig. 3b). In the transition event, the tunneling current first suddenly decreased towards a more negative value and then abruptly decreased. This reduction of the tunneling current corresponds to an increase of the tunneling gap.

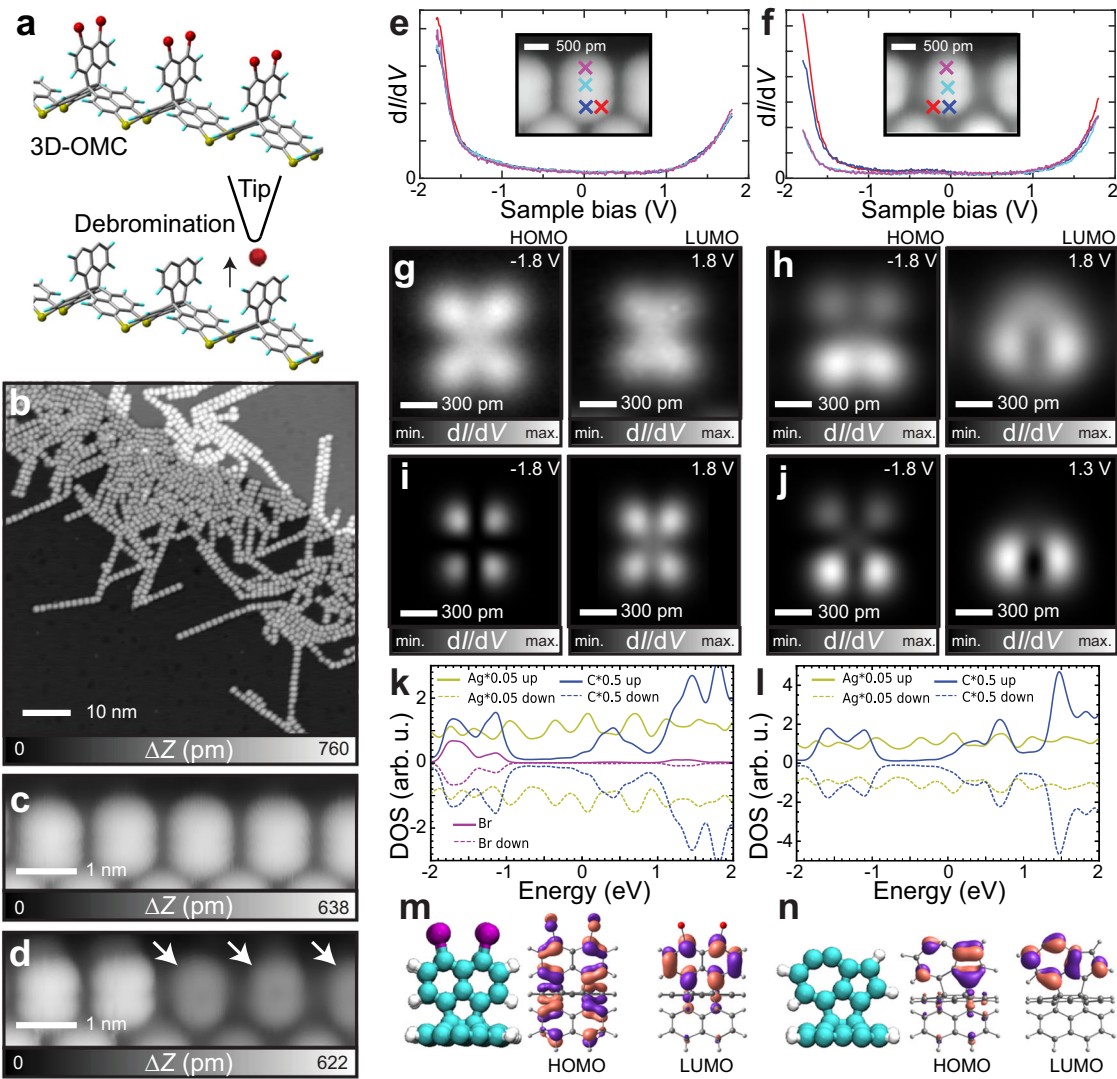

**Fig. 2 | On-surface synthesis of 3D-OMC and electronic properties of the unit before and after debromination. a** Schematic drawings of the three-dimensional organometallic compound (3D-OMC) and its diradical species. Red, blue and yellow balls in **c**, **d** correspond to Br, H, and Ag atoms. Grey lines correspond to C-C bonds, respectively. **b** Large-scale STM image of 3D-OMCs synthesized on Ag(111). **c** Close view before and **d** after tip-induced debromination. **e** d*I*/d*V* curves measured on the dibrominated naphthyl group **1** and **f** after the tip-induced debromination. The d*I*/d*V* curves were measured at the sites indicated by cross markers with the same colors in **e** and **f**. **g, h** d*I*/d*V* maps taken at LUMO and HOMO energy levels on **1** and

the debrominated unit. **i, j** Simulated constant height STM images at the labelled bias voltages and **k, l** the associated PDOS calculated from **m, n** the DFT optimized structures on five-layer Ag(111) surfaces on the left, and calculated HOMO and LUMO in vacuum (ROB3LYP-D3/6-311 G**) at the middle and right, respectively. Purple, green, white balls correspond to Br, C, and H, respectively. Purple and light red surfaces represent the relative signs of the orbital coefficients drawn at 0.04 e bohr$^{-3}$ level. Measurement parameters: Sample bias voltage $V = 500$ mV and tunneling current $I = 2$ pA in **b**, $V = 200$ mV and $I = 5$ pA in **c**, and $V = 1$ V and $I = 10$ pA in **d**.

Indeed, the initial bright spots darkened accompanied with an asymmetric contrast flip (Fig. 3c), indicating a successful chiral switch. We found that the switch can also be caused by applying a positive bias voltage of 2.2 V at the dark site as indicated by a red cross (Fig. 3c, e). This chiral switch (**3** $\iff$ **3'**) should occur through an energetically higher lying diradical intermediate **2**, which was the final product proposed before. To understand the manipulation mechanism, we calculated the energy barrier of the isomerization to the dehydroazulene unit **3** from the diradical unit **2** (Fig. 3f and Supplementary Note 1) and found that it is about 1.1 eV. Therefore, the isomerization should not occur without the influence of the tip at low temperature. The barrier height is also of an order that could be generally overcome by a combination of population of antibonding states weakening the bonds with a current heating due to inelastic electron tunneling, as we previously demonstrated[14]. This energy diagram also indicates that when the second debromination is induced, the molecular unit is

immediately transformed to the dehydroazulene, as we very rarely observed a symmetric contrast (Supplementary Fig. 5). To prove the high-reproducibility of the local probe isomerization, we created an embedding of 19 texts with the standard 8-bit binary ASCII code in the selected heptamer units, similar to the previous demonstration with several arrays of Fe atoms by Loth et al.[21]. The breadboard was prepared by sequential cleavage of C-Br bonds by applying high-bias voltages at chosen positions of the 3D-OMC (Supplementary Figure 6). Since the chirality of the isomers was initially random, we first reset all to be 0000 0000, namely "00" in the hexadecimal format. Then, the chirality of each unit was sequentially switched. Flipping four units, we embedded 0100 1110, which denotes "N". By conducting 71 subsequent manipulations, we typed "Nanoprobe GRP. NIMS©" (Fig. 3g), demonstrating the high controllability of the chiral in the dehydroazulene array by tip-induced isomerization (Supplementary Movie 1).

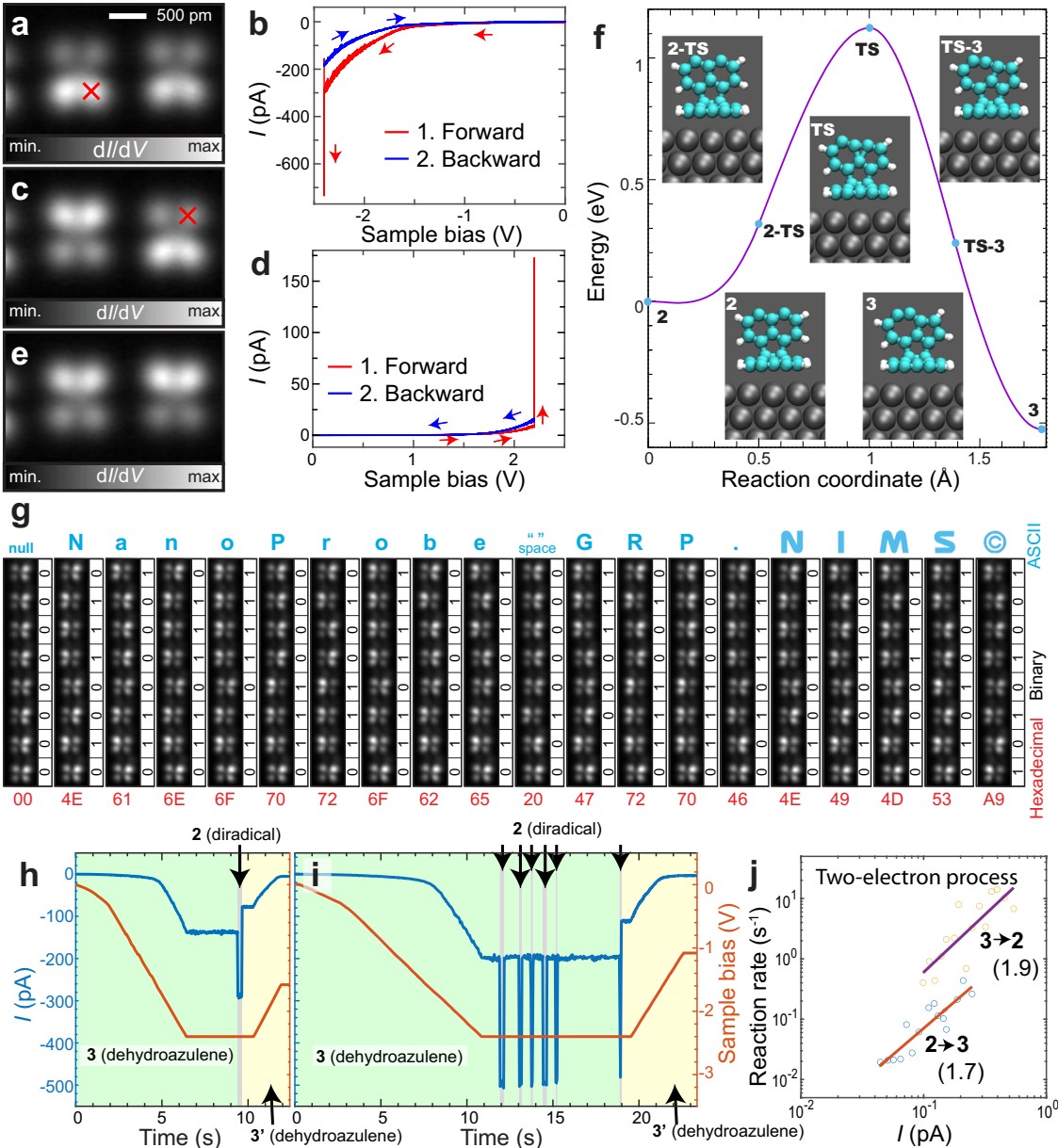

**Fig. 3 | Local probe-induced structural isomerization. a, c, e** STM topographies of dehydroazulene units **3** on a 3D-OMC. **b** *I-V* curve taken in negative and **d** positive voltage ranges. Abrupt changes of the tunneling current indicate the tip-induced isomerization of the dehydroazulene unit. **f** Calculated energy barrier for the structural isomerization. **g** Systematic local probe isomerization of dehydroazulene units on a 3D-OMC. "Nanoprobe GRP. NIMS©" in 8-bit binary ascii code is embedded via sequential 71 isomerization in the dehydroazulene array. The size of the image is 1.52 nm × 10.16 nm. **h, i** Tunneling current as a function of time measured at different tip-sample gaps. Green, gray, and yellow areas indicate the presence of initial dehydroazulene, short-lived diradical and final dehydroazulene units, respectively. **j** Reaction rates of azulene to diradical and diradical to azulene at different tunneling currents. The values of 1.9 and 1.7 indicate the slopes of linear fitting of each data in the log-log scale. Measurement parameters: $V = -1.8$ V and $V_{ac} = 10$ mV in **a, c, e**.

To investigate the reaction path, the tunneling current was recorded as a function of time during the tip-induced isomerization (**3** → **2** → **3'**, Fig. 3h). As the bias voltage was kept at −2.3 V, the tunneling current was constant until a sharp drop occurred, attributed to the presence of the diradical species **2**, which then quickly rebounds to a lesser value in magnitude of the tunneling current. This drop in the tunneling current after the rebound is an indication of the chiral switch, since the tunneling gap between the five-membered ring and the tip was larger than that between the seven-membered-ring and the tip. We also found transformation events back to the initial dehydroazulene. In Fig. 3i, for example, it is shown a case where the tunneling current rebounds 5 times to the same value of tunneling current, referring to the **2** → **3** event, until the chiral switch **2** → **3'** finally

occurs. The probability of reverse isomerization to the initial chirality for the diradical was higher than that to another chirality (289 events for **2** → **3** and 129 events for **2** → **3'**), indicating that the reaction barrier of **2** → **3'** is higher than that of **2** → **3** under the tip. The short-range interaction with the probe is most probably responsible for the preferential **2** → **3** isomerization. Nevertheless, such tip effects played a minor role in the reaction because once the chiral switch occurred, the probability to induce the reverse isomerization **3'** → **2** significantly reduced due to the drastic drop of the tunneling current. We recorded the tunneling current as a function of time during seven chirality switches at 15 different tip-sample distances and obtained mean reaction rates between dehydroazulene and diradical units at different mean tunneling currents (Fig. 3j). Although the reaction rates scattered

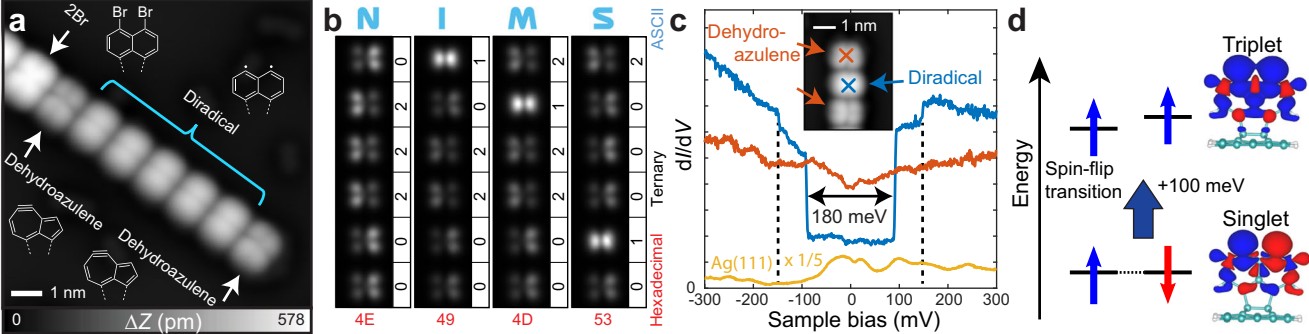

**Fig. 4 | Synthesis of a diradical molecular unit array and its electronic and magnetic properties. a** STM topography taken after tip-induced dehalogenation and subsequent isomerization to the diradical units. **b** Systematic local probe isomerization to embed "NIMS" in 5-bit ternary ascii code. The size of the image is 1.62 nm × 7.66 nm. **c** d$I$/d$V$ curves measured at the diradical and dehydroazulene units, as well as on Ag(111) for reference. **d** Schematic visualization of the inelastic spin flipping transition, and the DFT calculated spin densities of the open shell and triplet states of the diradical unit.

particularly at large-tip sample separations, we found that these isomerization processes were based on a two-electron process via the slopes of the fitted data[9], and that the reaction rate of $3 \rightarrow 2$ was approximately twelve times higher than that of $2 \rightarrow 3$. To try to understand this, we explored the charged states of the molecular units, using a fragment of the ribbon structure calculated at the B3LYP level[22] to represent the fact that any charging of the molecular units will be transient and rapidly transferred to the metallic substrate. These calculations showed (Supplementary Figure 7) that the diradical$^{-1}$ is 89 meV lower in energy than the dehydroazulene$^{-1}$, suggesting that electron attachment plays a role in the observed reaction rates.

**Diradical array and its electronic and magnetic properties**

We found that if the tip was set at a tunneling current gap below 100 pA with a sample bias of −2.4 V above the seven-membered ring of the dehydroazulene unit, the diradical usually remained stable for several tens of seconds. The lifetime was long enough to retract the tip and subsequently set the bias voltage to 0 V so that the diradical could be obtained as a product. By repeating this process, we obtained the diradical unit array in the 3D-OMC (Fig. 4a), which shows a symmetric contrast of the unit, in good agreement with the simulations (Supplementary Figure 4). The tip-induced isomerization among the diradical and the two dehydroazulenes units was also highly reproducible as we could embed "NIMS" in 6-bit ternary ascii code (0: low dehydroazulene, 1: diradical, 2: high dehydroazulene, Fig. 4b, Supplementary Movie 2). The radical was kept stable when the bias voltage was set in the range of −2 V to 2 V, (Supplementary Figure 8). We also measured the electronic properties of the diradical and the dehydroazulene units near the Fermi level (Fig. 4c and Supplementary Figure 9). Note that each curve is shifted for visibility. Although the differential conductance curve on the dehydroazulene unit is almost featureless, a distinct step-like feature is observed on the diradical unit, displaying a wide gap of 182 meV; which is likely related to an inelastic spin flipping transition of the unpaired electrons. Due to electron-electron interactions, a spin-spin coupling between localized electrons in each radical part arises. The value of such Heisenberg coupling depends on direct and superexchange contributions, leading to the appearance of a many-body excitation that can mediate electron tunneling. DFT calculations reveal that the ground state of the diradical displays spin-densities with opposite spins concentrated mostly in each radical part, configuring an open-shell singlet with anti-ferromagnetic (anti-FM) coupling between the unpaired electrons of the radical parts, while the triplet state with similar spin-density and unpaired electrons ferromagnetically (FM) coupled is observed 100 meV above the ground state. Total energies of a fragment of the ribbon structure containing the diradical part calculated at the B3LYP level shows that the triplet is 50 meV above the open shell singlet. The

previous findings show that the spin ground state of the molecule is likely a spin-singlet, where the exchange spin coupling accounts for the previous energy difference. An open-shell singlet as the ground state happens when no strong overlap is observed between the spin orbitals and has been observed in other diradical-organic molecules such as benzynes[23], for instance, and graphene fragments[24–26]. The spin density of anti-FM and FM configurations of the system plus the surface drawn at 0.001 e bohr$^{-3}$ using the VESTA software[27] are shown in Fig. 4d, while molecular orbitals and energy levels of a portion of the diradical calculated at the B3LYP level are shown in Supplementary Figure 10. The step-like conductance increase observed in Fig. 4c is associated with the inelastic excitation between the singlet and triplet configurations of the molecule as given by the Hamiltonian $H = 2J\mathbf{S}_1 \cdot \mathbf{S}_2$, with $\mathbf{S}_1$ and $\mathbf{S}_2$ being the spin operators in the two dangling orbitals, with $\mathbf{S}_i = (S_i^x, S_i^y, S_i^z)$. The energy scale $J$ of such a Hamiltonian can be extracted from first principles by calculating the energy difference between the anti-FM ($\uparrow\downarrow$) and FM ($\uparrow\uparrow$) configurations obtained by DFT. However, as the inelastic spin transition is given by the singlet-triplet $\frac{1}{\sqrt{2}}(\uparrow\downarrow - \downarrow\uparrow) \Rightarrow \uparrow\uparrow$ transition (the true eigenstates of the spin Hamiltonian), the step in the conductance appears at an energy $2J$ (Supplementary Note 2 and the Refs. [28,29]). Based on this, the estimated theoretical conductance inelastic step would appear at 100 meV, which is fairly close to the measured step 90 meV in the d$I$/d$V$ taken at the diradical unit. Another small step is observed around ±150 mV in the d$I$/d$V$ taken at the diradical unit, corresponding to a splitting of 60 meV. Since the anisotropy energy of the C atoms would be of less than 1 meV (as reference, the anisotropy energy for Co is 3 meV[30], Supplementary Figure 11), and given that the increase in conductance is much smaller than the spin-flipping transition step, we attribute this feature to inelastic tunneling due to vibrational modes of the molecule[31](Supplementary Notes 3 and 4). Further DFT calculations considering only a portion of the molecule that stands out from the ribbon show that several vibrational modes have the frequency compatible with the measured gap, and some of these modes have contributions from only atoms closer to the radical parts, as can be seen in Supplementary Table 1 and Supplementary Figure 12.

In summary, we present the synthesis of a dehydroazulene array in three-dimensional organometallic compounds via systematic tip-induced debromination and structural isomerization with scanning probe microscopy at low temperature under ultra-high vacuum. Two chiral dehydroazulene and the diradical configurations can be switched by applying bias voltages in a controlled manner and consequently 19 texts were embedded in 8-bit binary and 6-bit ternary ascii code via successful a number of tip-induced local reactions. Furthermore, we found that the diradical moiety hosts an open-shell singlet that can undergo an inelastic spin transition from antiferromagnetic to ferromagnetic coupling. Expanding the previous local chemical

reaction by which unprecedented planar molecules were synthesized and analyzed at the single-molecule level, we are now able to control the structures of the units in the molecular array. In addition to the tip-induced stereo isomerization[32], such systematic isomerization is of importance to advance nanochemistry towards fabrication of molecular systems molecule by molecule.

## Methods

### Experimental

All the experiments were conducted in a low temperature scanning tunneling microscopy (STM) system (homemade) at 4.3 K under an ultrahigh-vacuum environment ($<1 \times 10^{-10}$ mbar). The bias voltage was applied to the sample while the tip was electrically grounded. Au(111) and Ag(111) surfaces were cleaned through cyclic $Ar^+$ sputtering for 10 min and annealing at 720 K for 15 min. Hexabromo-substituted trinaphtho[3.3.3]propellane (6Br-TNP)[14] was deposited from Knudsen cells (Kentax GmbH). The impurity of 6Br-TNP was below the detection limit of $^1H$ and $^{13}C$ NMR measurements and was also confirmed by elemental analysis: Anal. Calcd for $C_{32}H_{12}Br_6$: C, 43.88; H, 1.38. Found: C, 44.00 H, 1.44. The samples were annealed at 370 K for 10 min to form the organometallic bonded compounds. The STM tip was made from a chemically etched tungsten wire. The modulation amplitude of the AC bias voltage was between 2 and 10 $mV_{0-peak}$ and the frequency was 510 Hz.

### Theoretical calculations

All first-principles calculations including the surface in this work were performed using the periodic plane-wave basis Vienna Ab initio simulation package (VASP) code[33,34] implementing the spin-polarized Density Functional Theory. To accurately include van der Waals interactions in this system, we used the DFT-D3 method with Becke-Jonson damping[35,36]. Projected augmented wave potentials were used to describe the core electrons[37] with a kinetic energy cutoff of 500 eV (with PREC = accurate). Systematic $k$-point convergence was checked for all systems with sampling chosen according to the system size. This approach converged the total energy of all the systems to the order of 1 meV. The properties of the bulk and surface of Ag were carefully checked within this methodology, and excellent agreement was achieved with experiments. For simplicity and easier comparison of electronic structure between isolated molecules and on-surface molecular structures, the metallic adatoms binding the OMC to the surface were neglected in simulations. Previous calculations[14] showed that there were minimal differences in the OMC and GNR molecular structures far from the substrate. This can be further seen in a direct comparison between the two in Supplementary Fig. 3. For calculations of the ribbons on the surface, a vacuum gap of at least 1.5 nm was used, and the size of the silver slab ($7 \times 6$ unit cells) and 3D-GNR length ($2 \times 1$) were chosen to minimize the lattice mismatch compared to the optimized structure of the unsupported 3D-GNR (4.6% for this combination). A $3 \times 3 \times 1$ $k$-point grid was used and the upper three layers of Ag (five layers in total) and all atoms in the ribbon were allowed to relax to a force of less than 0.01 eV Å$^{-1}$. Atomic structure visualizations were made with the VMD package[38]. Simulated STM images were calculated using the CRITIC2 package[39,40] based on the Tersoff−Hamann approximation[41]. An equivalent set of calculations for the azulene structure on the Au surface was performed, but we observed no significant differences with respect to the results on Ag. Barrier calculations were performed initially using the standard Nudged Elastic Band (NEB) method with increasing image density[42], before implementing the Climbing NEB approach[43] to find the final barrier. For these calculations, only the gamma point was used. Additional calculations of molecular orbitals and vibrational frequencies at the B3LYP level[22] were realized with the ORCA code[44] using the def2-TZVP basis set[45] or Gaussian-16 program (revision B.01)[46] with the 6-311G** basis

set. We also performed Climbing NEB calculations of the neutral and anionic systems by manually controlling the total charge of the system, which was either 0 or -1, and all possible spin multiplicities were taken into consideration. In all calculations performed with ORCA, we considered only fragments of the structure that stand out from the ribbon, but appreciate all the local arrangements of the small aromatic compounds that form the diradical and the dehydroazulene.

## Data availability

The data that support the findings of this study are available from the corresponding author upon request. The simulated data is available via Zenodo[47].

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

## Acknowledgements

This work was supported in part by the Japan Society for the Promotion of Science (JSPS) KAKENHI Grant Number JP22H00285, and the Academy of Finland project 346824. Computing resources from the Aalto Science-IT project and CSC, Helsinki, are gratefully acknowledged. A.S.F. has been supported by the World Premier International Research Center Initiative (WPI), MEXT, Japan. We thank Ondrej Krejčí for useful discussions.

## Author contributions

S.K. planned and conducted experiments. S.K., Z.Y., K.S. and O.C. analyzed and discussed the experimental data. T.N., T. Ko. and T. Ku. designed and synthesized the precursor molecule. O.S., L.K., J.L.L. and A.S.F. conducted theoretical calculations. S.K. and A.S.F. contributed to writing the manuscript. All authors commented on the manuscript.

## Competing interests

The authors declare no competing interests.
