## [Peer Review File · Nature Communications]

Local Probe-induced Structural Isomerization in a One-Dimensional Molecular ArrayEditorial Note: This manuscript has been previously reviewed at another journal that is not operating a transparent peer review scheme. This document only contains reviewer comments and rebuttal letters for versions considered at Nature Communications.

REVIEWERS' COMMENTS

Reviewer #1 (Remarks to the Author):

I thank the authors for the detailed responses to clarify the referee's concerns. The manuscript has been improved; therefore, I recommend publication in Nature Communications.

Reviewer #2 (Remarks to the Author):

Following the comments, suggestions and remarks provided by the previous referee reports, the authors have significantly improved their manuscript. I recommend publishing the manuscript.

Reviewer #3 (Remarks to the Author):

In the revised manuscript, the authors have clearly addressed the comments raised by the reviewers. I am therefore pleased to recommend its publication in Nature Communications.

I have a few minor suggestions.

-In the article since Figure 3E and F have been removed, then the information regarding the two figures should be removed from the figure caption.

In the Supplementary file:

- On page S2 of the SI: I suggest to add "ULTRA"-high vacuum in line 30, which is more appropriate for the low range of pressures indicated.

- In Figure S2, the authors should check if the position of the yellow line III-VI is correct since it is placed on the substrate, far from the debrominated molecular unit.

- On page S11 the triple gap formula is not readable (line 172).

Reviewer #4 (Remarks to the Author):

I think the paper in its new version has improved significantly. The authors took their time and answered not only my questions but also the question of the 3 other referees.

I think the paper is now ready to be published at Nature Communications.

Referee #1 (Remarks to the Author):

I thank the authors for the detailed responses to clarify the referee's concerns. The manuscript has been improved; therefore, I recommend publication in Nature Communications.

→

We thank Referee#1 for supporting our work.

Referee #2 (Remarks to the Author):

Following the comments, suggestions and remarks provided by the previous referee reports, the authors have significantly improved their manuscript. I recommend publishing the manuscript.

→

We thank Referee #2 for supporting our work.

Referee #3 (Remarks to the Author):

In the revised manuscript, the authors have clearly addressed the comments raised by the reviewers. I am therefore pleased to recommend its publication in Nature Communications.

→

We thank Referee #3 for supporting our work and giving us suggestions to improve our manuscript.

I have a few minor suggestions.

-In the article since Figure 3E and F have been removed, then the information regarding the two figures should be removed from the figure caption.

→

We removed the information from the figure caption.

- On page S2 of the SI: I suggest to add “ULTRA”-high vacuum in line 30, which is more appropriate for the low range of pressures indicated

→

We added “ultra”, which is now described in method section in the main text.

- In Figure S2, the authors should check if the position of the yellow line III-VI is correct since it is placed on the substrate, far from the debrominated molecular unit.

→

The position is correct. Since the radical unit does not have bromine atoms, the height of the unit is lower than that of the brominated unit. To explain this, we added “Constant-height” in the figure caption.

- On page S11 the triple gap formula is not readable (line 172).

→

We inserted the formula again.

Referee #4 (Remarks to the Author):

I think the paper in its new version has improved significantly. The authors took their time and answered not only my questions but also the question of the 3 other referees.

→

We thank Referee #4 for supporting our work.